# MiniFold:
# Simple, Fast, and Accurate Protein Structure Prediction

**Jeremy Wohlwend**[*]                                                   *jwohlwend@csail.mit.edu*
*MIT*

**Mateo Reveiz**[*]                                                         *mreveiz@mit.edu*
*MIT*

**Matt McPartlon**                                                    *mmcpartlon@uchicago.edu*
*University of Chicago*

**Axel Feldmann**                                                         *axelf@csail.mit.edu*
*MIT*

**Wengong Jin**                                                       *wengong@csail.mit.edu*
*Broad Institute of MIT and Harvard*

**Regina Barzilay**                                                     *regina@csail.mit.edu*
*MIT*

**Reviewed on OpenReview:** *https://openreview.net/forum?id=1p9hQTbjgo*

## Abstract

Protein structure prediction has emerged as a powerful tool for biologists and drug makers. However, the computational cost of state-of-the-art models such as AlphaFold limits their scalability and makes training and fine-tuning prohibitively expensive. Although previous work has achieved considerable inference speedups by replacing the multiple sequence alignment step with protein language models, the overall architecture of structure prediction models, inherited from AlphaFold2, has remained largely unchanged. In this work, we show that protein language model-based structure predictors can be dramatically simplified at little to no loss in accuracy. Our model, MiniFold, consists of a redesigned Evoformer and a lightweight structure module. We also propose two novel GPU kernels, tailored to the proposed architecture. Equipped with the same ESM2 protein language model, MiniFold is competitive with ESMFold on the standard CAMEO and CASP datasets while achieving training and inference speedups of up to 20x, and significant reductions in peak memory. Our results show that MiniFold is an effective solution for large-scale applications and resource-constrained environments. Our code and trained models are available at https://github.com/jwohlwend/minifold.

## 1 Introduction

With advances in deep learning-based protein structure prediction, models such as Alphafold Jumper et al. (2021) are now routinely used in biological discovery. These models take as input a protein sequence and predict the 3-D coordinates of every atom in the protein. Yet, the significant computational cost associated with these tools limits their use in large scale applications. As an illustration, consider the task of identifying disease-relevant antibodies from patients' blood. A sequenced antibody repertoire may contain tens of

---

* Joint first authors

millions of unique clonotypes (i.e unique protein sequences) Briney et al. (2019). At a speed of just a few seconds per sequence, scanning 10 millions sequences would require over a GPU year. In addition to virtual screening, other applications, such as mutational scans, downstream fine-tuning, and the filtering of *de novo* designs, where ESMFold is broadly utilized Campbell et al. (2024); Watson et al. (2023), would all benefit from more efficient models. Importantly, efficient architectures can also accelerate the research life cycle in a space that typically requires extensive computational resources.

Previous works have already recognized this challenge. ESMFold Lin et al. (2023) and OmegaFold Wu et al. (2022) propose to replace the compute-heavy multiple sequence alignment (MSA) stage of AlphaFold with a protein language model (PLM), while conserving most of the original folding trunk design. By bypassing MSA generation, the time required for structure prediction can be reduced from minutes to seconds per sequence. However, for the various use cases outlined above, the cost and throughput of high-quality PLM-based folding models remain severely limiting, not to mention the largely unchanged cost of model training. We hypothesized that further improvements to these models could be achieved by addressing the bottlenecks inherited from AlphaFold. With the ESMFold architecture as a starting point, our goal was to create a more efficient model by pinpointing critical performance components and reducing computational bottlenecks.

Despite its considerable number of parameters, the protein language model used in ESMFold only covers a small percentage of the inference time ($< 5\%$). Instead, the bulk of the compute is spent in the Evoformer blocks ($\sim 80\%$), dominated by triangular operations, and to a lesser degree in the structure module ($\sim 15\%$), dominated by the invariant point attention (IPA). Our goal, therefore, is to design an efficient protein structure prediction model that addresses these two bottlenecks. Of note, the concurrently published AlphaFold3 and Boltz-1 models retain most of the original Evoformer design Abramson et al. (2024); Wohlwend et al. (2024), making optimization of this architecture highly desirable.

We present MiniFold, an efficient architecture for protein structure prediction which reduces an Evoformer block to a single bidirectional triangular multiplicative operation and feed-forward layers, and replaces the structure module with a lightweight transformer with pairwise bias. In addition, we introduce a hardware-optimized implementation using newly devised GPU kernels to enhance both the throughput and memory efficiency of the model. Specifically, our proposed kernel optimizations are applied to the triangular multiplicative updates and to the feed-forward layers in the revised Evoformer (Miniformer). We train MiniFold using the same PLM used in ESMFold on a single node of 8x A100 GPU's, and achieve competitive performance on the standard CAMEO and CASP test sets Kryshtafovych et al. (2021; 2023). When compared against the original architecture, MiniFold achieves 20x speedup and a significant reduction in peak memory utilization. Our work further highlights the potential for PLMs to not only accelerate homology search but also simplify downstream architectures, thereby enabling high-throughput protein structure prediction.

## 2 Related Works

**Protein structure prediction**    The field of protein structure prediction has seen significant advancements due to recent breakthroughs in deep learning methods. The current state-of-the-art model AlphaFold Jumper et al. (2021); Abramson et al. (2024) consists of two main components: a multiple sequence alignment (MSA) and template search, and a folding module responsible for decoding the MSA into a set of coordinates in 3-dimensional space. The derived MSA and templates are used to construct an initial set of sequence-level and pairwise-level representations. These are then fed to the folding module, more specifically to a pairwise track which learns to predict the pairwise distances between amino acid residues as a distogram. Finally, a structure module uses these pairwise embeddings to predict the 3D coordinates of each residue. RosettaFold Baek et al. (2021) and UniFold Li et al. (2022) adopt a similar MSA-based workflow but with different architectures. An important limitation of AlphaFold2, however, is its computational cost. Recently, protein folding models, such as OmegaFold Wu et al. (2022) and ESMFold Lin et al. (2023), accelerate inference by replacing the MSA database search with protein language models Rives et al. (2021); Meier et al. (2021); Lin et al. (2023); Elnaggar et al. (2023); Chen et al. (2023). The input to these models requires only single sequences, which is ideal for orphan proteins that do not have MSA. In contrast from PLM-based models that keep the same architecture as AlphaFold2, we focus on improving the efficiency of the structure prediction model to further improve its scalability. Concurrent to this work, the new AlphaFold3 and Boltz-1 models retain most of

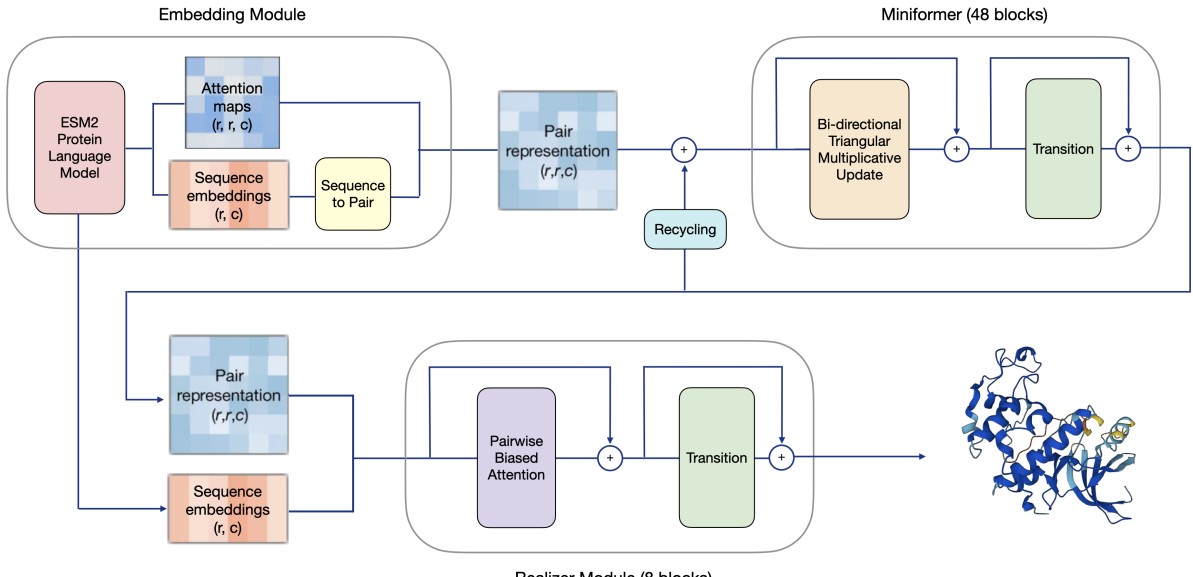

Figure 1: Our proposed MiniFold architecture. We use the ESM-2 protein language model to extract sequence level embeddings as well as pairwise attention maps. These are concatenated and fed to the Miniformer module which updates the pairwise embeddings using efficient triangular multiplicative update operations and feed-forward layers. Finally, a structure realizer composed of a gated transformer produces the full-atom coordinates from the pairwise and sequence embeddings.

the original Evoformer, both in the MSA module and in the similar Pairformer Abramson et al. (2024); Wohlwend et al. (2024), which remains very relevant to our work.

**Efficiency & scalability** Strategies to improve the efficiency of neural networks typically fall under one of four categories: pruning, quantization, GPU kernel optimization, and improvements in algorithmic complexity. Each of these approaches have been extensively studied in the context of the Transformer architecture Vaswani et al. (2017). One such example is the recently proposed Flash-attention Dao et al. (2022), which provides major speedup and memory savings to attention based architecture using an IO-aware implementation. Another example is Simple Recurrent Unit Lei (2021), which uses various GPU kernel optimization techniques for acceleration. Katharopoulos et al. (2020); Wang et al. (2020) propose algorithmic alterations that result in linear time complexity as a function of the input size. Yet, these methods are not obviously applicable to protein folding models, which utilize unique neural layers and transformations. Recognizing the importance of customized solutions to the model architecture at hand, our work aims to optimize protein structure prediction models to accelerate their inference speed, and improve their capacity to scale to long protein sequences. While recent work Cheng et al. (2022); Wang et al. (2022) has begun to explore hardware-level optimization techniques to accelerate protein folding training, these are limited by the current architecture. Chowdhury et al. (2022) propose a faster architecture in RGN2 but require refinement through Alphafold2 to reach competitive performance, which negates the potential speed-up. In contrast, our work focuses on both architecture-level and custom hardware-level optimizations to improve speed and memory utilization while retaining accuracy.

## 3 Methods

Our MiniFold architecture is composed of three modules: a sequence embedding module, an efficient Evoformer which we name Miniformer, and a lightweight structure module based on gated self-attention. Figure 1 illustrates the forward pass of our model. First, we encode the input protein sequence using the three-billion

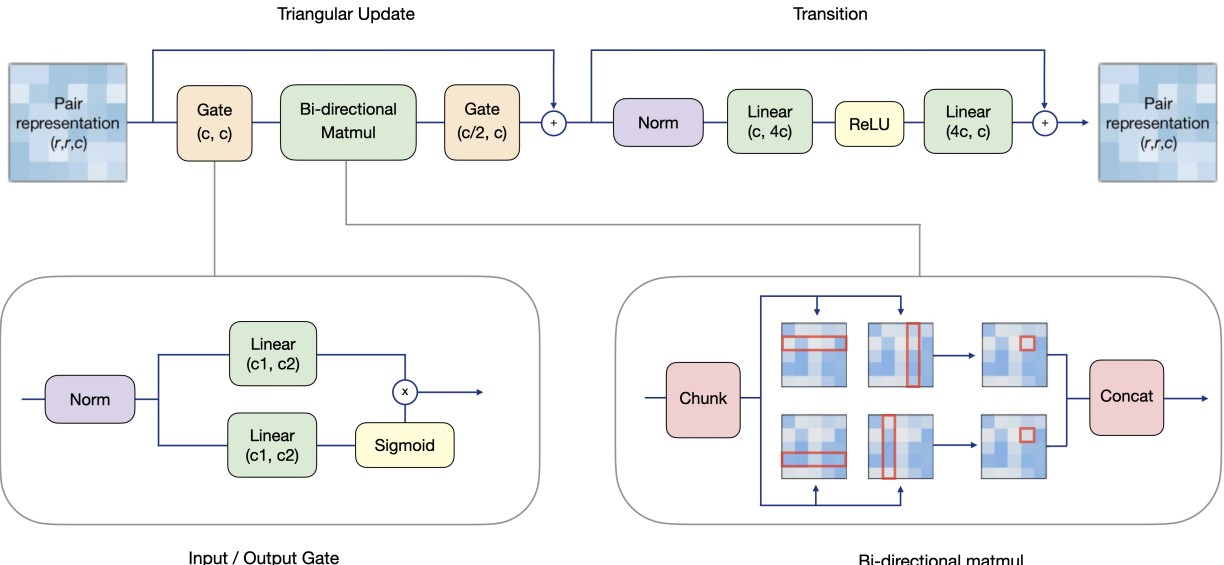

Figure 2: Our proposed Miniformer block. The input is a pairwise tensor of dimension $(r, r, c)$. We first compute the input gate as displayed in the bottom left panel. In the bi-directional matmul, we first chunk the tensor along the channel dimension to create 4 sub-tensors of shape $(r, r, c/4)$, and perform a pair of tensor products, once over the vanilla tensors and once over their transpose. The two resulting tensors of shape $(r, r, c/4)$ are then concatenated and fed to the output gate. This is followed by a skip connection and a two layer feed-forward network with a ReLU activation.

parameter ESM-2 language model Lin et al. (2023), which is a pre-trained 36 layer Transformer. The embeddings at the last layer of ESM-2 are fed into a small feed-forward network and tiled into pairwise representations such that every entry $(i, j)$ is the concatenation of the vector representations of residues $i$ and $j$. In addition, we also feed the concatenation of all the network's attention maps, and fine-tune the last 2 layers of the protein language model, both of which accelerate convergence during the early stages of training. This pair representation is then updated through 12 or 48 Miniformer blocks, where we propose the use of a redesigned bi-directional triangular multiplicative layer. Finally, the output of the Miniformer and the sequence embeddings from the protein language model are passed to a structure realizer which produces the 3D coordinates. In contrast to previous work, we apply recycling over the pairwise blocks only. The efficiency of our approach comes from the redesigned Miniformer blocks and the lighter structure realizer. In the following sections, we provide a detailed overview of these design choices and their motivation.

## 3.1 From Evoformer to Miniformer

We propose several modifications to the Evoformer architecture. Our new design is shown in Algorithm 1. First, we eliminate the sequence track and keep only the pairwise representation and update. While this change does not have a substantial impact on speed, it dramatically reduces the number of parameters in the Evoformer. Specifically, the sequence track in ESMFold holds over 600M parameters. When removed, less than 25M parameters remain. Through ablating this change, we argue that the representational capacity of the Evoformer is influenced by the depth and complexity of its operations rather than by its parameter count.

Next, we eliminate the Triangular Attention blocks. There are two reasons for this change. First, this operation produces attention maps that encode every node triplet, which results in substantial memory consumption that scales poorly with sequence length. Secondly, early in our experiments, we found that the expressive power of the Evoformer was driven by the triangular multiplicative blocks and not the triangular

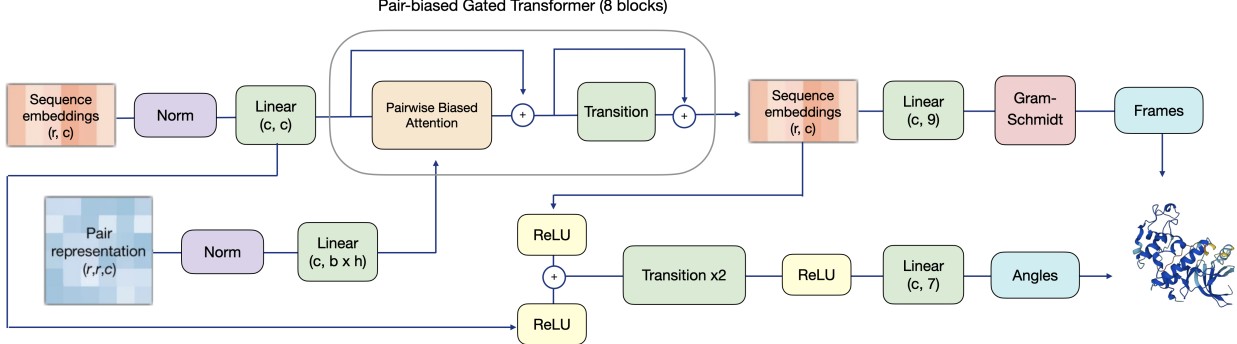

Figure 3: Our proposed transformer-based realizer. Pairwise embeddings are projected to an $(r, r, b \cdot h)$ tensor, where $b$ is the number of blocks and $h$ the number of attention heads. We compute 8 blocks of biased transformer layers, and the final sequence embeddings are used to predict a frame at each residue via Gram-Schmidt and the angles via the Angle Resnet described in AlphaFold2.

attention. This is ideal because although the complexity of the multiplicative update is also cubic in sequence length, the space complexity remains quadratic, resulting in a much cheaper operation in practice. Another potential advantage of this change would be to scale up other parts of the model, such as the pairwise embedding dimension, or the number of blocks. We leave this exploration for future work.

Finally, we propose a redesigned triangular multiplicative layer. We merge the incoming and outgoing blocks, fusing the point operations across the two blocks and removing one of the skip connections. We use a down-projection of the input dimension prior to the channel-wise matrix multiplication. We also modify the output gating to be a function of the matrix multiplication output instead of a function of the layer's input. This final change unifies the input and output gating operations and allows for better GPU kernel design, as explained in section 2.3. We refer to this layer as a bi-directional triangular update, which we combine with a feed-forward network in each Miniformer block as shown in Algorithm 1.

## 3.2 From Structure Module to Realizer

The structure module in AlphaFold2 takes as input the pairwise representation from the Evoformer and outputs the full atom 3D coordinates. We sought to investigate the role of the structure module, hypothesizing that most of the predictive capacity of the model already occurred in the Evoformer. In particular, for the large scale applications mentioned such as virtual screening, it may not be necessary to compute the 3D coordinates. Pairwise distances may be sufficient to identify stable folds, and discriminate for binders. Yet, in order to substantially alter or omit the structure module, we would first need to ensure that its role is primarily to realize the 3D coordinates that match the predicted distogram. Motivated by this hypothesis, we hereby propose two alternatives for the structure module: a parameter-free multidimensional scaling realizer (MDS) Mead (1992), and a fast transformer realizer.

**Parameter-free MDS realizer** In order to validate our hypothesis we first attempted to directly obtain the coordinates from the predicted distogram, using MDS. To do so, we first determine the predicted distance for each pair of residues by choosing the distance bin with the highest probability. Since the maximum distance bin is 25Å, we fill the missing entries (i.e., pairs predicted to be farther than 25Å) by defining a graph with nodes as $C\alpha$ atoms and edges as distances under 25Å and run the all-pair shortest path algorithm on this graph. This results in an approximate distance for missing entries. Given the approximated full distance matrix, we then perform classical MDS to generate initial coordinates for the atoms. Because MDS is sensitive to noise, the resulting coordinates generally do not satisfy the predicted distogram. Therefore, we refine the coordinates through 10 iterations of stress majorization with an LBFGS optimizer. We provide pseudo-code for this approach in algorithm 3. In the absence of a structure module, we also propose a

change to the recycling strategy by directly recycling the argmax of the distogram to the Miniformer. More specifically, we convert the maximum likelihood distance bin into a one-hot representation which we linearly project and sum into the input to the Miniformer blocks. Only $C\alpha$ atoms are considered in this setting.*

**Transformer-based realizer**  While the MDS approach serves as a strong indicator of the role of the structure module, the cost of running shortest path, MDS and gradient descent is substantial. This is not necessarily an issue in a setting where pairwise distances are sufficient but can be a limitation when the 3D coordinates must be computed. We therefore thought to replace it by learning to perform MDS using a neural network that is similar but more efficient than the existing IPA-based structure module. To achieve this, we propose a transformer with biased attention, similar to what is used in the sequence track in the original Evoformer 2. Starting with the initial pairwise representation $z$ and the sequence embeddings $s$, we first project $z$ into a set of attention biases, with one bias matrix per transformer head. This bias is added to the attention scores prior to the softmax. In contrast to the IPA, we do not iteratively update coordinates in each block, but instead predict the 3D frames of each residue in one-shot after the final layer. To do so, we found it helpful to construct the frame by predicting the coordinates of the $N$, $C\alpha$ and $C$ atoms and applying Gram-Schmidt, instead of directly predicting a translation and quaternion at each position. In line with casting the structure module as a realizer, the idea of only applying recycling in the Miniformer blocks was kept to further improve training and inference speed. We predict side-chain atoms using 5 torsion angles. All angles are predicted by their cosine and sine contributions as per Alphafold2, using feed-forward layers on top of the sequence representation of the transformer.

### 3.3  Efficient GPU Inference Kernels

Recent work has shown that hardware-optimized implementations can yield substantial speedups by fusing memory intensive operations into a single GPU kernel. Motivated by this success, we identified two promising areas of improvement in the Miniformer architecture: the self-gating operation in the multiplicative update, and the feed-forward layers where the hidden projection quadruples the input dimension. The main idea is to fuse operations to avoid excessive memory transfers. We implement these new kernels using the Triton library Tillet et al. (2019), which provides block-level control over memory transfers while abstracting away the details of issuing tensor core operations.

**Self-gating**  The following gating operation appears twice in each triangular update:

$$\mathbf{y} = (\mathbf{Vz} + \mathbf{b_1}) \cdot \text{sigmoid}(\mathbf{Wz} + \mathbf{b_2})$$

Because all of these operations are applied entry-wise, and $z$ is loaded twice, speedup and memory savings could be obtained by fusing this operation in a single kernel. The key insight is that the hidden dimension of $z$ is small enough (i.e 128) that the GPU SRAM can partition $z$ across its large input dimension only. As such, we can reduce the number of reads and the number of intermediate writes saving both memory and computing time. We provide pseudo-code in Algorithm 4 in the appendix.

**Feed-forward**  The feed-forward layer is ubiquitous in model deep learning architectures. It consists of an initial linear projection, followed by a non-linear activation (in our case a ReLU), and another linear projection. This layer is an essential part of the Transformer architecture Vaswani et al. (2017), as well as the folding blocks described above. This layer typically uses a hidden dimension that is larger than the input and output dimensions. In our case, it is 4x the input dimension. Our goal here is to avoid materializing the inner matrix, which can causes a substantial memory bottleneck:

$$\mathbf{y} = (\mathbf{W_2}\text{ReLU}(\mathbf{W_1z} + \mathbf{b_1}) + \mathbf{b_2})$$

Here, again we observe that GPU SRAM is sufficiently large to partition z across its large input dimension. Each thread block loads several complete rows of $z$ into SRAM and is responsible for producing several

---

* A similar MDS approach was independently developed by Sergey Ovchinnikov and presented at a talk during CASP14.

| Dataset | Model | TMscore | RMSD | lDDT | RMSD-aa | lDDT-aa | Runtime (s/seq) |
|---|---|---|---|---|---|---|---|
| CAMEO | MiniFold-12 | 0.825 | 3.220 | 0.837 | 3.946 | 0.764 | 0.359 |
| | MiniFold-48 | 0.846 | 2.781 | 0.859 | 3.479 | 0.786 | 0.807 |
| | ESMFold | 0.837 | 2.888 | 0.856 | 3.611 | 0.784 | 11.402 |
| | OmegaFold | 0.815 | 3.382 | 0.837 | 4.111 | 0.767 | 61.507 |
| | AlphaFold2 | 0.880 | 2.100 | 0.905 | 2.763 | 0.828 | 77.842 |
| | RosettaFold1 | 0.761 | 4.659 | 0.798 | 4.661 | 0.795 | 227.368 |
| | RosettaFold2 | 0.883 | 1.987 | 0.906 | 2.666 | 0.823 | 14.269 |
| CASP14 | MiniFold-12 | 0.616 | 7.126 | 0.663 | 7.827 | 0.608 | 0.531 |
| | MiniFold-48 | 0.641 | 6.452 | 0.692 | 7.154 | 0.636 | 1.390 |
| | ESMFold | 0.656 | 6.100 | 0.699 | 6.850 | 0.638 | 12.844 |
| | OmegaFold | 0.706 | 5.366 | 0.737 | 6.186 | 0.673 | 149.303 |
| | AlphaFold2 | 0.831 | 3.002 | 0.849 | 3.698 | 0.772 | 105.374 |
| | RosettaFold1 | 0.634 | 8.174 | 0.683 | 8.145 | 0.682 | 310.294 |
| | RosettaFold2 | 0.798 | 3.489 | 0.826 | 4.307 | 0.746 | 18.176 |
| CASP15 | MiniFold-12 | 0.617 | 9.424 | 0.703 | 10.393 | 0.650 | 0.510 |
| | MiniFold-48 | 0.657 | 8.174 | 0.736 | 9.035 | 0.679 | 1.312 |
| | ESMFold | 0.659 | 8.172 | 0.755 | 9.116 | 0.688 | 10.484 |
| | OmegaFold | 0.631 | 10.080 | 0.731 | 11.271 | 0.667 | 128.875 |
| | AlphaFold2 | 0.728 | 5.754 | 0.819 | 6.740 | 0.748 | 107.173 |
| | RosettaFold1 | 0.588 | 11.378 | 0.683 | 11.362 | 0.711 | 321.987 |
| | RosettaFold2 | 0.706 | 6.441 | 0.798 | 7.589 | 0.727 | 19.786 |

Table 1: Performance of MiniFold (12 and 48 layers) and baselines on the CAMEO, CASP14 and CASP15 datasets. We report the TMScore, RMSD, lDDT, RMSD All-atoms (RMSD-aa) and lDDT All-atoms (lDDT-aa). Each metric is computed by taking the mean across the targets for each dataset, except for the RMSD and RMSD All-atoms metrics where we use the geometric mean. The average runtime in seconds / sequence is provided for each model and dataset. Runtimes exclude the generation of multiple sequence alignments (MSA) for MSA-based models.

complete rows of y. This means that every thread block can loop over the expanded dimension of W1 without materializing any intermediates in HBM. Instead we load $z$ once and proceed to loop over the output dimension of $W_1$ and accumulate the partial matrix multiplication of $W_1 z$ into the output $y$. This strategy results in memory usage that is only a function of the input and output, but not the inner representation. We provide pseudo-code in Algorithm 5 in the appendix.

**Layer-norm** We observe that both kernels are always preceded by a Layer normalization. To further reduce intermediate data transfers, we fuse the layer norm into the respective kernels. Unless stated otherwise, our kernel results include the fusing of the layer norm.

### 3.4 Training objective

AlphaFold and ESMFold were trained using a combination of multiple objective functions including distogram loss, FAPE loss, structural violation loss, confidence loss, etc. Similar to previous work, we construct equally sized bins ranging from 2 to 25 Angstroms, and train the model to predict the pairwise distances between each pair of residues, using a classification objective over the bins described above, with the common cross entropy loss. We initially train using only this objective, and without enabling recyling or the sturcture realizer. After this initial training stage of the Miniformer, we begin a second training stage where we enable both the structure realizer and recycling. The model is supervised using the backbone and side-chain FAPE losses, and the supervised chi losses as previously done in Alphafold2. We also train a pLDDT predictor on top of the structure realizer with the same loss used in AlphaFold2. The model is trained for about 250K steps during each training stage.

| Dataset | MiniFold-12 | MiniFold-12 MDS | MiniFold-48 | MiniFold-48 MDS |
|---------|-------------|-----------------|-------------|-----------------|
| CAMEO  | 0.88 / 0.14 | 0.88 / 0.15 | 0.9 / 0.13 | 0.9 / 0.14 |
| CASP14 | 0.73 / 0.22 | 0.74 / 0.24 | 0.77 / 0.22 | 0.78 / 0.25 |
| CASP15 | 0.8 / 0.23 | 0.8 / 0.24 | 0.84 / 0.23 | 0.84 / 0.23 |
| CAMEO  | 2.75 / 7.63 | 2.88 / 22.05 | 2.34 / 7.2 | 2.42 / 21.83 |
| CASP14 | 6.24 / 13.75 | 11.34 / 23.31 | 6.19 / 13.6 | 8.22 / 21.98 |
| CASP15 | 6.9 / 32.31 | 11.05 / 54.99 | 6.73 / 33.06 | 8.11 / 40.74 |

Table 2: $C\alpha$ lDDT (top) and $C\alpha$ RMSD (bottom) comparison between the lightweight structure module and the MDS realizer on CAMEO, CASP14 and CASP15 datasets, for MiniFold with 12 and 48 miniFormer blocks. Values correspond to median/sample standard deviation (n-1 normalization).

## 4 Experiments

We showcase the efficacy of our approach by comparing the structure prediction accuracy, inference speed, and memory consumption of MiniFold against ESMFold in Table 1. We also provide comparisons against Omegafold, and several MSA-based models: AlphaFold2, RosettaFold1, RosettaFold2 Jumper et al. (2021); Baek et al. (2021); Lin et al. (2023); Wu et al. (2022). All models were run with 3 recycling rounds (for a total of 4 forward passes), and the same MSA precomputed using the ColabFold service was used for all MSA-based models Mirdita et al. (2022). Our detailed experimental setting can be found in the Appendix E and validation curves can be found in appendix F.

### 4.1 Prediction Accuracy

#### 4.1.1 Results

**Accuracy**  We report the performance of MiniFold in Table 1, and some example predictions in Figure C. Our results show that MiniFold is competitive with other protein language model-based structure prediction models, matching ESMFold on the CAMEO and CASP15 datasets, at a fraction of the computational cost. MiniFold is also competitive, though slightly weaker on CASP14. Of note, the 12 layer model is surprisingly strong, only underperforming the full-sized model by a few lDDT points. MiniFold outperforms OmegaFold on the CAMEO and CASP15 datasets. Of note, Appendix G shows that MiniFold and ESMFold both perform comparably at different depth of sequence homologs, including orphan proteins with little to no homologs.

As a limitation, the performance of both ESMFold and MiniFold remains weaker than AlphaFold2 and RosettaFold2 on the CASP datasets. This is an expected result, which can be largely attributed to the language model being less effective than MSA retrieval in extracting evolutionary context Lin et al. (2023). While future work may consider training an MSA-based version of MiniFold that can be more directly compared to these tools, we do not claim that all our observations are directly applicable to MSA based models such as AlphaFold2. Instead, our results highlight that the architecture of pLM based structure prediction models need not necessarily be the same as their MSA counterpart. Since MiniFold, similarly to ESMFold, trails the performance of MSA based models, it may be beneficial to combine the speed of MiniFold with the accuracy of a more computationally expensive method by applying them in succession to achieve a good balance in a large scale screen.

We also report a comparison between the MDS and transformer-based structure realizers in Table 2. Surprisingly, we find that the non-parametric MDS approach is just as effective as the transformer model in recovering the $C\alpha$ trace, though a small regression in RMSD is noticeable. Although using MDS is impractical because of its lengthy computation time, this result suggests that the underlying structure has already been discovered by the pairwise track and that distogram supervision alone can be effective in predicting protein structure. It also helps put into perspective the surprising effectiveness of a simple transformer com-

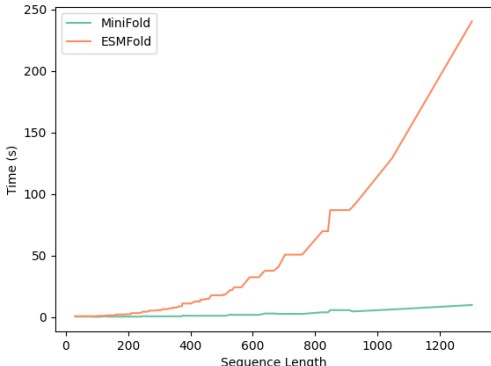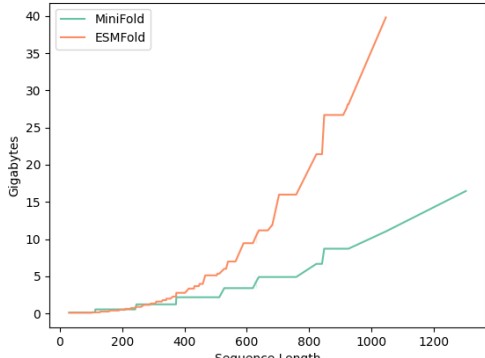

Figure 4: Left panel: overall compute time as a function of the sequence length, computed over the 191 targets in the CAMEO test set. MiniFold is 10 to 20x faster and shows improved scaling as a function of the input sequence length. Right panel: peak memory usage at different sequence lengths.

pared to equivariant architecture such as the IPA. Interestingly, AlphaFold3 reached a similar conclusion, opting for a non-equivariant structure module.

**Uncertainty estimation** Current protein folding models provide pLDDT scores to quantify the uncertainty of predicted structures. We also train a pLDDT predictor, which obtains a correlation of 0.76 on CAMEO and 0.9 on CASP14. In addition, we demonstrate that the entropy of the predicted distogram is a reasonable indicator of uncertainty. As shown in Figure 7, we find that predictions with lower entropy (e.g., higher certainty) tend to have higher LDDT scores. The Pearson correlation between true LDDT and distogram entropy is 0.60 on the CAMEO dataset and 0.9 on the CASP14 dataset. Although not as effective as the pLDDT predictor, the distogram entropy is visibly a powerful measure of uncertainty.

## 4.2 MiniFold Efficiency

### 4.2.1 Experimental Setting

We perform a systematic analysis of the throughput and memory usage of our proposed MiniFold architecture. We run MiniFold, ESMFold and OpenFold's implementation of AlphaFold2 on the CAMEO test set, and compute the throughput at different sequence lengths. Furthermore, we analyze the step-by-step progression from the Evoformer used in the ESMFold model to our proposed Miniformer:

- **Full**: This is the original Evoformer baseline in the ESMFold implementation, which uses the triangular updates from the OpenFold project Ahdritz et al. (2022). We test the full Evoformer has 48 folding blocks and 3 recycling steps.

- **No sequence**: We remove the sequence track, keeping only the pairwise track

- **No attention**: We remove the triangular attention, but keep the triangular update.

- **Bi-directional**: We fuse the outgoing and incoming layers and include the down-projection.

- **Ours**: We include our two optimized GPU kernels in the final Miniformer.

### 4.2.2 Results

As shown in Figure 4, When measuring speed end to end by running each tool over the CAMEO test set, we observe considerable speedup over ESMFold and the reference OpenFold implementation(ignoring MSA

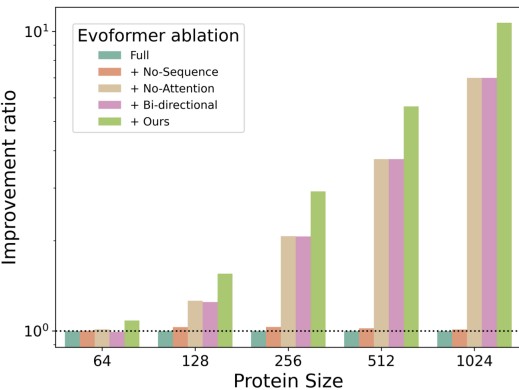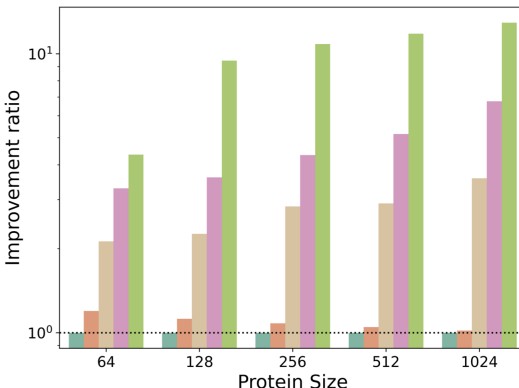

Figure 5: Ablations for Miniformer memory (left) and time (right) improvement ratio. The full model (cyan bar) corresponds to the original Evoformer implementation in ESMFold. Our Miniformer model (green bar) with all optimization techniques (Removing sequence track and triangular attention, adding kernels, bi-directional) achieves 10-20x improvement in throughput over Evoformer. The model results in significant speedup and memory efficiency depending on the sequence length.

compute time), with speedups ranging from 10x for a length 400 protein up to 20x for a length 1000 protein. We also report a controlled ablation on our proposed Miniformer.

Our results show that each of the steps proposed above contribute favorably to the model's improved efficiency. We note that the removal of the sequence track has minimal effect on inference speed, but reduces the number of trainable parameters considerably (from 600M down to 23M in ESMFold). The removal of the triangular attention and the use of a downward projection in the multiplicative update yield substantial speedups, achieving nearly 5x improvement over the baseline. Our proposed kernels provide an another 2 to 3x speedup, leading to a total of 10 to 20x improvement in throughput for the full size model, with improved scaling as a function of protein length.

The bottom panel of Figure 5 measures the peak memory usage during inference over different protein lengths. Here, we see that our proposed kernels result in substantial savings, and that MiniFold improves memory efficiency by nearly 10x for longer proteins. These results have important consequences regarding inference as well as training, allowing larger batch sizes which can result in faster training times.

## 5  Conclusion

In this work, we propose a highly efficient architecture for protein structure prediction, and a hardware-optimized implementation that results in considerable savings in both speed and memory while conserving most of its expressive power and performance. Our results have important implications regarding the use of protein structure prediction models for high throughput applications. On the other hand, we observe that MiniFold, similar to ESMFold, has room for improvement before matching the results of Alphafold2 and the all atom capabilities of RoseTTAFold All-Atom Krishna et al. (2024), AlphaFold3 Abramson et al. (2024) and Boltz-1 Wohlwend et al. (2024). While our work builds from the success of PLM's, future work could aim to bridge the performance gap by expanding MiniFold to the use of multiple sequence alignments (MSA) as input, and to explore methods to speed up the construction and utilization of these MSAs. The methods described here could also be generalized to multi-chain complexes as well as full atom prediction. As such, this work serves as a stepping stone towards the goal of matching AlphaFold3 and Boltz's performance at a fraction of the computational cost.

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

## A   Architecture details

Algorithm 1 outlines the architecture of the miniFormer block, which takes as initial input the pairwise representations $z \in R^{L \times L \times c}$ from the embedding module and updates it through triangular multiplicative updates and transition layers.

---

**Algorithm 1** Miniformer Block, related to Figure 2

---

Triangular($\{z_{ij}\}, c = 128$) :

1 : $z_{ij} \leftarrow \text{LayerNorm}(z_{ij})$        $z_{ij} \in R^c$

2 : $a_{ij}, b_{ij}, c_{ij}, d_{ij} \leftarrow \text{sigmoid}(\text{Linear}(z_{ij})) \cdot \text{Linear}(z_{ij})$        $a_{ij}, b_{ij}, c_{ij}, d_{ij} \in R^{c/4}$

3 : $z_{ij} \leftarrow \text{Concat}\left(\sum_k a_{ik} \cdot b_{jk}, \ \sum_k c_{ki} \cdot d_{kj}\right)$        $z_{ij} \in R^{c/2}$

4 : $z_{ij} \leftarrow \text{LayerNorm}(z_{ij})$        $z_{ij} \in R^{c/2}$

5 : $z_{ij} \leftarrow \text{sigmoid}(\text{Linear}(z_{ij})) \cdot \text{Linear}(z_{ij})$        $z_{ij} \in R^c$

Transition($\{z_{ij}\}, c = 128$) :

1 : $z_{ij} \leftarrow \text{LayerNorm}(z_{ij})$        $z_{ij} \in R^c$

2 : $z_{ij} \leftarrow \text{ReLU}(\text{Linear}(z_{ij}))$        $z_{ij} \in R^{4c}$

3 : $z_{ij} \leftarrow \text{Linear}(z_{ij})$        $z_{ij} \in R^c$

MiniformerBlock($\{z_{ij}\}, c = 128$) :

1 : $z_{ij} \leftarrow z_{ij} + \text{Triangular}(z_{ij})$        $z_{ij} \in R^c$

2 : $z_{ij} \leftarrow z_{ij} + \text{Transition}(z_{ij})$        $z_{ij} \in R^c$

---

Algorithm 2 outlines the architecture of the transformer-based realizer, which replaces the IPA-based structure module found in AlphaFold 2. The Gram-Schmidt, AngleResnet and ComputeAllAtomCoordinates subroutines are as described in AlphaFold 2 Jumper et al. (2021). The algorithm takes as input the single and pair representations $s \in R^{L \times c_s}$ and $z \in R^{L \times L \times c_s}$

---

**Algorithm 2** Transformer-based realizer, related to Figure 3

---

TransformerRealizer($\{s_i\}, \{z_{ij}\}, N_l = 8, H = 16, c_s = 1024, c_z = 128$) :

1 : $s_i^{initial} \leftarrow \text{LayerNorm}(s_i^{inital})$        $s_i \in R^{c_s}$

2 : $s_i \leftarrow \text{Linear}(s_i^{inital})$

3 : $z_{ij} \leftarrow \text{LayerNorm}(z_{ij})$        $z_{ij} \in R^{c_z}$

4 : $b_{ij}^{N_l h} \leftarrow \text{Linear}(z_{ij})$        $b_{ij}^{N_l h} \in R^{N_l H}$

5 : $\text{For}(l \in [1, ..., N_l])$

6 :     $\{s_i\} \leftarrow \{s_i\} + \text{PairBiasedSelAttn}_l(\{s_i\}, b_{ij}^l)$

7 :     $\{s_i\} \leftarrow \{s_i\} + \text{Transition}_l(\{s_i\})$

\# Get backbone frames

8 : $N_i, Ca_i, C_i = \text{Linear}(s_i)$

9 : $T_i = \text{Gram-Schmidt}(N_i, Ca_i, C_i)$

\# Proceed as in AlphaFold2 Structure module

10 : $\alpha_i^f = \text{AngleResnet}(s_i^{inital}, s_i)$

11 : $T_i^f, x_i^a = ComputeAllAtomCoordinates(\alpha_i^f, T_i)$

12 : $\text{Return}(T_i^f, x_i^a, \{s_i\}, T_i, \alpha_i^f)$

---

Algorithm 3 outlines the procedure used to realize the distogram coordinates with a paramter-free method. The ShortestPath subroutine uses some variation of the Floyd-Warshall algorithm.

---

**Algorithm 3** Coordinate Realizer

---

**def** MDS(logits $\in \mathbb{R}^{N \times N \times 64}, d_{max} = 25\text{Å}, e \in \mathbb{R}^{64}$) :

1 : $D_{ij} \leftarrow \sum_{b=1}^{64} argmax((logits_{ij})_b \cdot e - b)$

2 : $D_{ii} \leftarrow 0$

3 : $W_{ij} \leftarrow \begin{cases} 1, & \text{if } (D_{ij} < d_{max}) \ \& \ (i \neq j) \\ 0, & \text{otherwise} \end{cases}$

4 : $\mathbf{D} \leftarrow \text{ShortestPath}(\mathbf{D} \odot \mathbf{W})$

5 : $J_{ij} \leftarrow \begin{cases} \frac{1}{N}, & \text{if } (i \neq j) \\ 0, & \text{otherwise} \end{cases}$

6 : $\mathbf{B} \leftarrow -\frac{1}{2} * \mathbf{J} * \mathbf{D}^{\circ 2} * \mathbf{J}$

7 : $\Lambda, \mathbf{V} \leftarrow \text{eigen-decomposition}(\mathbf{B})$

8 : $\mathbf{C}_\alpha \leftarrow \mathbf{V} \odot \sqrt{\Lambda}$

9: for $i \in [1, 2, 3]$

10 : $\quad \mathbf{C}_\alpha \leftarrow \text{LBFGS}(\mathbf{D}, \mathbf{C}_\alpha)$

11 : Return $\mathbf{C}_\alpha$ $\hfill \mathbf{C}_\alpha \in \mathbb{R}^{N \times 3}$

---

## B    Kernels

### B.1    Algorithms

The following pseudocode outlines the Triton kernels implemented during model inference for the self-gating and feed-forward subroutines. Our kernels show large improvements, even when compared to torch.compile.

---

**Algorithm 4** Self-gating kernel. Each parallel kernel thread $i$ is responsible for rows $R_i$ and columns $C_i$ of the output matrix $Y$. Computes $Y = \text{Linear}(V, X) \cdot \text{Sigmoid}(\text{Linear}(W, X))$

---

**def** SelfGating(X, V, W, Y, i, B):

1 : $X_i \leftarrow X[R_i, :]$ $\hfill$ Incurs one read operation for $X$

2 : $V_i \leftarrow V[:, C_i]$

3 : $W_i \leftarrow W[:, C_i]$

4 : $Y_i^1 \leftarrow DOT(X_i, V_i)$

5 : $Y_i^2 \leftarrow DOT(X_i, W_i)$

6 : $Y_i \leftarrow \text{MUL}(\text{SIGMOID}(Y_i^1), Y_i^2)$

7 : $Y[R_i, C_i] \leftarrow Y_i$ $\hfill$ Incurs one write operation for $Y$

---

---

**Algorithm 5** Feed-forward kernel. Each parallel kernel thread $i$ is responsible for rows $R_i$ and all columns of the output matrix $Y$. Computes $Y = X + \text{Linear}(\text{ReLU}(\text{Linear}(X, V)), W)$

---

**def** FFN(X, V, W, Y, i, B):

1: $X_i \leftarrow X[R_i, :])$                                             Incurs one read operation for $X$

2: for $j \in [1, ..., N//B]$

3:    $V_j \leftarrow V[:, C_j]$                                   Here $C_j$ is a block of B columns of V

4:    $W_j \leftarrow W[C_j, :]$                                   $C_j$ is also a block of B rows of W

5:    $H \leftarrow \text{RELU}(\text{DOT}(X_i, V_j))$

6:    $X_i \leftarrow X_i + \text{DOT}(H, W_j)$

7: $Y[R_i, :] \leftarrow X_i$                                      Incurs one write operation for $Y$

## B.2 Latency measurements

| Size | Torch | Torch-compile | Ours |
|------|-------|---------------|------|
| 64.0 | 0.125465 | 0.139198 | 0.217369 |
| 128.0 | 0.074751 | 0.136619 | 0.218885 |
| 256.0 | 0.263739 | 0.135980 | 0.214086 |
| 512.0 | 1.004088 | 0.442307 | 0.218418 |
| 1024.0 | 3.891327 | 1.666970 | 0.868068 |
| 2048.0 | 18.806410 | 6.639621 | 3.469150 |

Table 3: Self-gating kernel latency (ms)

| Size | Torch | Torch-compile | Ours |
|------|-------|---------------|------|
| 64.0 | 0.047169 | 0.142224 | 0.126898 |
| 128.0 | 0.120064 | 0.135038 | 0.124615 |
| 256.0 | 0.403863 | 0.373916 | 0.180016 |
| 512.0 | 1.543569 | 1.366344 | 0.463024 |
| 1024.0 | 6.119958 | 5.369287 | 1.726875 |
| 2048.0 | 32.731358 | 23.447216 | 6.805521 |

Table 4: MLP kernel latency (ms)

## C Predictions

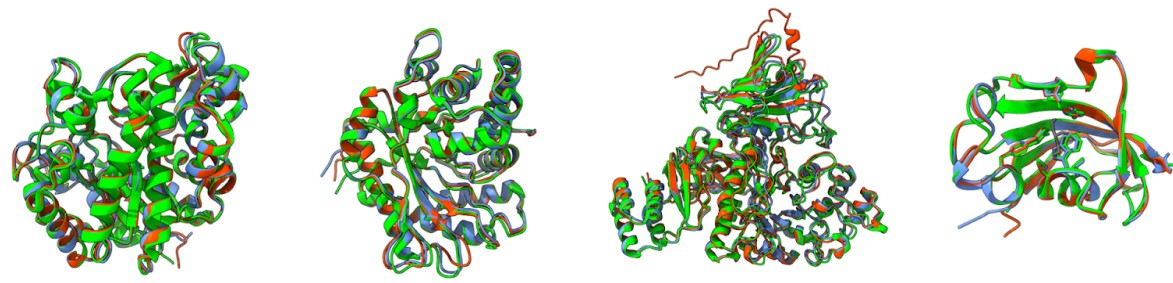

Figure 6: Predicted structures for MiniFold (blue), ESMFold (red) and the ground truth PDB structure (green). PDB ID from left to right: 7e5j, 7efs, 7rcz, 7dki.

## D   Uncertainty prediction

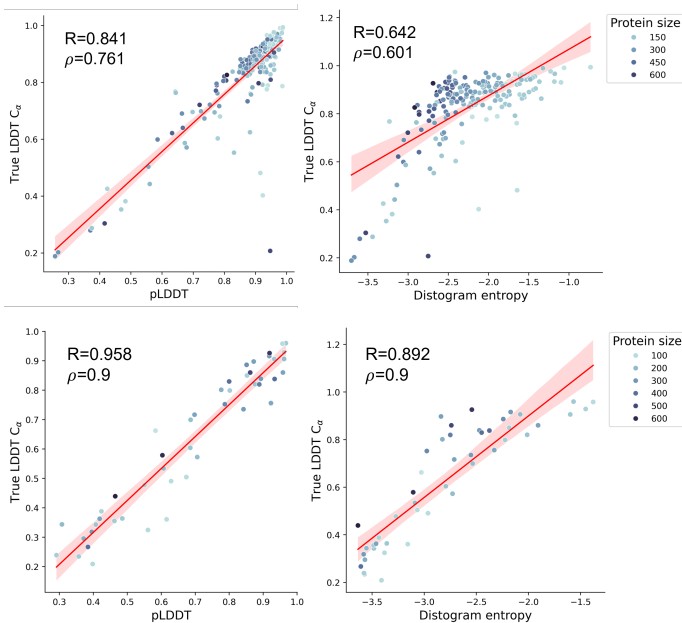

Figure 7: The trained pLDDT predictor (left) and the entropy over the predicted distogram (right) correlate with the true lDDT $C_\alpha$ as evaluated in the CAMEO dataset. CAMEO on top block and CASP14 on bottom block. The R value was computed using the scipy.stats.linregress function and the spearman correlation $\rho$ was computed with stats.spearmanr with default parameters in python.

## E   Experimental Setting

**Training and validation set**   Similarly to ESMFold, we constructed a training set from the AlphaFold Protein Structure database. Varadi et al. (2023). We first cluster Uniref50 sequences Suzek et al. (2007) at 30% sequence similarity, and then select the structures with an average pLDDT score above 0.7. This results in a high quality, diverse dataset of roughly 10 million structures. Interestingly, we found that training on this distillation set alone was sufficient to achieve high accuracy. We randomly sample 10000 structure as our fixed validation set for model selection and hyper-parameter optimization.

**Benchmarks**   We evaluate MiniFold using CAMEO targets Haas et al. (2019) released between April 1, 2022 and June 31, 2022, for a total of 191 structures. Similar to ESMFold, we also evaluate on 51 structures from the CASP14 targets, chosen to avoid overlaps with the ESMFold training set. We also use 47 targets from CASP15 with structures made recently publicly available. The specific protein ID's used for each dataset can be found in appendix H.

**Metrics**   We report the local distance difference test (lDDT) and root mean square deviation (RMSD) for each dataset and model. We also report both full-atom metrics. A higher metric correspond to better performance for lDDT, and lower is better for RMSD. When computing metrics we align the reference sequence to the residues available in the structure using the USalign tool Zhang et al. (2022). We also ignore ambiguous residues (X, U, B, Z) in the evaluation, as well as any missing atom in the target structure.

**Hyper-parameters and training details**   We train MiniFold using 12 and 48 Miniformer blocks. We train the models on a single node using 8x A100 GPU's and a batch size of 16 per GPU for an effective batch size of 128 similar to ESMFold. We use the Adam optimizer with a learning rate of 3e-5 for the tuned PLM layers, 1e-4 for the structure module parameters and 1e-3 for all others for all parameters. We use a

pairwise embedding dimension of 128 in the Miniformer, and a sequence embedding dimension of 1024 with 16 self-attention heads each of 64 dimensions in the structure realizer. Training is performed in two stages: first by limiting to the embedder and miniFormer blocks and then by enabling recycling and the structure realizer.

## F  Validation Curve

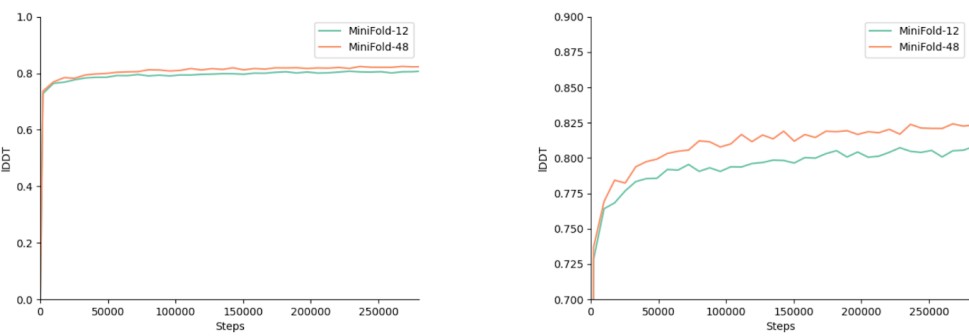

Figure 8: Validation curve as a function of training steps. Left panel: full scale, Right panel: zoomed-in.

## G  MSA Depth

We show the performance of ESMFold and MiniFold as a function of MSA depth, showing that MiniFold performs comparably well on orphan proteins with shallow to no sequence homologs. Both models show degradation when less evolutionary signal is available as was previously reported by Lin et al. (2023). This figure was constructed by comparing the performance of the model against the depth of the MSA produced by the colabfold MSA server on these sequences.

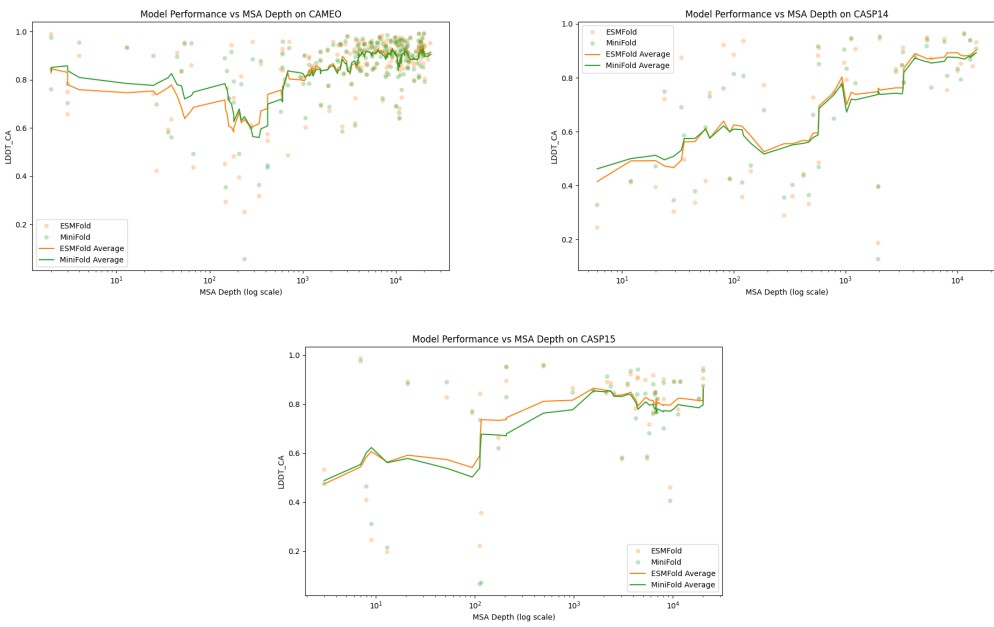

Figure 9: LDDT-$C\alpha$ as a function of MSA depth (i.e known sequence homologs).

# H  Targets

The following list contains the target id and corresponding chain (if applicable) used for evaluation.

## H.1  CAMEO

7W74-A, 7PC6-A, 7L8N-A, 7VW0-B, 7E4S-A, 7QDV-A, 7K2Z-A, 7F6J-C, 7E1C-A, 7EA4-A, 7WKQ-A, 7PW1-A, 7EL8-A, 7X8V-A, 7U5Y-A, 7NTN-A, 7O9F-B, 7BGS-B, 7RCZ-A, 7WWR-A, 7TJ1-D, 7VOH-A, 7KO9-A, 7LGR-A, 7LVE-A, 7STT-A, 7QDW-A, 7ESO-A, 7EKZ-A, 7RXE-A, 7PUO-A, 7QU2-A, 7E1B-A, 7E52-A, 7E8K-B, 7VU7-A, 7VNO-A, 7ETR-A, 7TOC-A, 7TLF-I, 7QBG-E, 7WEW-A, 7ED1-A, 7ACY-B, 7RCW-A, 7VMC-B, 7BI4-A, 7EFS-D, 7E4J-A, 7WIN-B, 7TLF-G, 7TA5-A, 7PMO-G, 7EAD-A, 7KW6-A, 7Q51-A, 7OPY-F, 7S8T-J, 7MYV-B, 7S94-C, 7TKV-A, 7TA9-A, 7LQN-A, 7M5W-A, 7KPJ-E, 7RGV-A, 7EGT-B, 7MO6-B, 7E0V-C, 7E3T-A, 7P0H-A, 7O4O-A, 7U5F-D, 7BEW-A, 7NDE-A, 7P3I-B, 7THH-A, 7QBZ-A, 7BLL-A, 6Y0D-A, 7VU5-A, 7LH6-A, 7KDX-B, 7FIW-B, 7TBU-A, 7S2R-B, 7RW4-A, 7LK4-D, 7MKK-B, 7PHW-E, 7U4H-A, 7TT9-A, 7E04-A, 7LVF-A, 7PRD-A, 7A67-A, 7LI0-A, 7QSU-A, 7S5C-A, 7WJ9-A, 7T1Y-C, 7W1F-B, 7PUJ-A, 7R74-B, 7EHG-E, 7WGK-A, 7D66-C, 7VEE-A, 7MCC-A, 7EBQ-A, 7LSV-B, 7THW-A, 7SGN-C, 7LQM-D, 7ETR-C, 7F6L-B, 7A67-B, 7V9F-A, 7N40-C, 7LXK-A, 7QBP-A, 7V4S-A, 7F8A-A, 7JIZ-A, 7PUG-A, 7PZJ-A, 7QSS-A, 7WNW-B, 7DKI-A, 7JT9-A, 7KOB-A, 7N0E-A, 7Q05-E, 7E0L-A, 7MHU-A, 7B7T-A, 7TB5-A, 7WRK-A, 7VWT-A, 7LXS-A, 7VSP-C, 7BBZ-A, 7EBT-B, 7ON9-A, 7AAL-A, 7V8E-B, 7B9P-A, 7WRP-A, 7F3A-A, 7OS0-A, 7E4U-A, 7ERV-A, 7W26-A, 7SKC-A, 6ZPP-A, 7TXP-A, 7NUV-A, 7DUP-A, 7V9H-A, 7EYM-A, 7MWR-A, 7STZ-C, 7WWX-A, 7ED6-A, 7V6I-A, 7EWU-B, 7BCB-B, 7E5J-A, 7TCR-C, 7REJ-A, 7ERP-B, 7PAB-A, 7R6P-A, 7EZG-A, 7E8J-A, 7TV9-C, 7OM4-B, 7NMI-B, 7E3Q-A, 7T03-A, 7FFA-A, 7VJS-A, 7EYL-A, 7R09-A, 7TMU-C, 7B0K-A, 7AB3-E, 7DT1-A, 7N40-B, 7RY7-A, 7EJG-C

## H.2  CASP14

T1024, T1025, T1026, T1027, T1028, T1029, T1030, T1031, T1032, T1033, T1034, T1035, T1036s1, T1037, T1038, T1039, T1040, T1041, T1042, T1043, T1044, T1045s1, T1045s2, T1046s1, T1046s2, T1047s1, T1047s2, T1049, T1050, T1053, T1054, T1055, T1056, T1057, T1058, T1064, T1065s1, T1065s2, T1067, T1070, T1073, T1074, T1076, T1078, T1079, T1080, T1082, T1089, T1090, T1091, T1099

## H.3  CASP15

T1104, T1106s1, T1106s2, T1112, T1113, T1114s1, T1114s2, T1114s3, T1119, T1120, T1121, T1122, T1123, T1124, T1125, T1132, T1133, T1134s1, T1134s2, T1137s1, T1137s2, T1137s3, T1137s4, T1137s5, T1137s6, T1137s7, T1137s8, T1137s9, T1145, T1147, T1151s2, T1152, T1154, T1158, T1159, T1170, T1173, T1174, T1176, T1178, T1179, T1185s1, T1185s2, T1185s4, T1187, T1188, T1194

