# OpenReview forum: "MiniFold: Simple, Fast, and Accurate Protein Structure Prediction"
_TMLR — Accepted by TMLR_

### Review · Reviewer_AYru · 2025-02-18

**Summary Of Contributions:**

The paper introduces MiniFold, a simplified and computationally efficient architecture for protein structure prediction, aiming to address the scalability and resource limitations of existing models such as AlphaFold2 and ESMFold. By redesigning the Evoformer and structure module, and introducing optimized GPU kernels, the authors claim significant improvements in efficiency (up to 20x speedup) with minimal loss in accuracy. The architecture is evaluated on standard datasets (CAMEO, CASP14, CASP15), demonstrating competitive performance with existing methods while achieving drastic reductions in runtime and memory usage.

**Audience:**

Yes

**Claims And Evidence:**

Yes

**Requested Changes:**

**1. Address Accuracy-Performance Trade-off**
Provide more analysis or discussion on the scenarios where MiniFold’s reduced accuracy might limit its use, and suggest potential solutions (e.g., hybrid models that combine MSAs and PLMs).

**2. Evaluate on More Datasets**
Testing MiniFold on additional benchmarks or datasets, particularly those with multi-chain complexes or orphan proteins, would provide stronger evidence of its generalizability.

**Strengths And Weaknesses:**

**Stengths**
1. The authors propose a simplified architecture (MiniFold) that reduces computational cost while maintaining competitive accuracy. The reported speedups (10x-20x) and memory savings are impressive and make MiniFold a strong candidate for large-scale applications like virtual screening, mutational scans, and de novo protein design.
2. MiniFold is benchmarked against state-of-the-art models on widely recognized datasets (CAMEO, CASP14, CASP15). The results are well-documented, showing that MiniFold performs comparably to ESMFold while being significantly faster.

**Weaknesses**
1. While MiniFold achieves significant efficiency gains, its accuracy (e.g., lDDT scores) lags behind AlphaFold2 and RosettaFold2 on CASP datasets. This limitation is acknowledged but could be emphasized more clearly in the discussion section, especially for applications requiring high precision.
2. The authors mention that MiniFold could be extended to MSA-based inputs to improve accuracy, but no concrete experiments or evidence supporting this claim are presented.

---

> ### Author Response · Authors · 2025-03-07
> **Author response**
>
> We thank the reviewer for their thoughtful comments. Please see our response to your points below.
>
> > While MiniFold achieves significant efficiency gains, its accuracy (e.g., lDDT scores) lags behind AlphaFold2 and RosettaFold2 on CASP datasets. This limitation is acknowledged but could be emphasized more clearly in the discussion section, especially for applications requiring high precision.
>
> In many applications that require large scale screening, it is acceptable to trade off speed for some accuracy loss. This is evident from the broad use of ESMFold in the field, for example in filtering de-novo protein designs where efficiency is highly desirable [1,2]. The main point of comparison for our method is ESMFold.  ESMFold and AlphaFold2 use the same architecture, with the exception of the MSA input which is replaced by pLM features. Here, since we take ESMFold as our base model to optimize, the same drop in performance with respect to Alphafold2 is expected [3]. However, the key result is that using the very same pLM as ESMFold, we can match ESMFold’s performance with a much more streamlined architecture, reducing cost in the many settings where ESMFold is currently utilized. This being said, we do agree that there are settings where the highest accuracy is necessary at which point other approaches may be preferable. We have emphasized this point by making modifications to the result section (4.1.1) and the conclusion.
>
> > The authors mention that MiniFold could be extended to MSA-based inputs to improve accuracy, but no concrete experiments or evidence supporting this claim are presented
>
> We do mention that this is a possible area for future work, but do not claim that these results are readily applicable to MSA based models and have modified the text in the result section that lacked clarity on this point (see the updated section 4.1.1 and the conclusion).
>
> With regards to specific requested changes:
>
> > 1. Address Accuracy-Performance Trade-off Provide more analysis or discussion on the scenarios where MiniFold’s reduced accuracy might limit its use, and suggest potential solutions (e.g., hybrid models that combine MSAs and PLMs)
>
> We thank the reviewer for this suggestion. We have extended the discussion on this point, by adding a paragraph at the end of section 4.1.1.
>
> > 2. Evaluate on More Datasets Testing MiniFold on additional benchmarks or datasets, particularly those with multi-chain complexes or orphan proteins, would provide stronger evidence of its generalizability.
>
> As a single chain protein model, MiniFold is not intended to be used on multi-chain complexes. Regarding orphan proteins, we have added a figure in appendix G showing the model performance as a function of MSA depth for the different datasets, which shows that our model is competitive with ESMFold in all regimes of homology depth, including orphan proteins with little to no homologs.
>
> [1] Campbell, A., Yim, J., Barzilay, R., Rainforth, T. &amp; Jaakkola, T.. (2024). Generative Flows on Discrete State-Spaces: Enabling Multimodal Flows with Applications to Protein Co-Design. Proceedings of the 41st International Conference on Machine Learning
>
> [2] Watson, J.L., Juergens, D., Bennett, N.R. et al. De novo design of protein structure and function with RFdiffusion. Nature 620, 1089–1100 (2023). https://doi.org/10.1038/s41586-023-06415-8
>
> [3] Zeming Lin et al. ,Evolutionary-scale prediction of atomic-level protein structure with a language model.Science379,1123-1130(2023).DOI:10.1126/science.ade2574

---

> > ### Comment · Reviewer_AYru · 2025-03-25
> >
> > Thanks for your response. Overall, the acceleration achieved is not only substantial but also indicative of strong scalability potential.

---

### Review · Reviewer_iLp6 · 2025-02-18

**Summary Of Contributions:**

The paper introduces MiniFold, an efficient approach to protein structure prediction that simplifies the complex Evoformer architecture from models like AlphaFold2. By eliminating the sequence channel and triangular attention in favor of a bi-directional triangular multiplicative update, and by leveraging a pre-trained protein language model (ESM-2) to bypass the computational cost of generating multiple sequence alignments (MSA)—the authors achieve significant improvements in both speed and memory efficiency. Additionally, custom GPU kernel optimizations using Triton further enhance the model’s performance, making MiniFold a promising candidate for high-throughput and resource-constrained applications.

**Audience:**

Yes

**Claims And Evidence:**

Yes

**Requested Changes:**

Refer to weaknesses.

**Strengths And Weaknesses:**

Strengths:
1. Efficiency and Resource Savings:
The paper introduces optimizations by simplifying the Evoformer architecture: removing the sequence channel and triangular attention, and retaining only the bi-directional triangular multiplicative update. These improvements are particularly valuable for large-scale protein structure prediction and virtual screening tasks.

2. Practical MSA Replacement:
By leveraging a pre-trained protein language model (ESM-2) to extract sequence embeddings, MiniFold avoids the computationally expensive generation of MSA. This is especially useful for orphan proteins or scenarios where rich evolutionary data is unavailable, making the method more broadly applicable.

Weaknesses:

1. Accuracy Trade-offs:
Although MiniFold performs well on many datasets, its accuracy, particularly on the CASP14 dataset and in full-atom structure prediction, is inferior to that of more mature methods such as AlphaFold2 or RosettaFold2. This suggests that the architectural simplifications might compromise the ability to capture fine-grained structural details in challenging cases.

2. Dependence on Pre-trained PLMs:
While replacing MSA with a pre-trained language model saves significant time, it may also limit performance in cases where deep evolutionary information (typically captured via MSA) plays a critical role in accurate structure prediction. This reliance might result in diminished performance for proteins where evolutionary context is paramount.

3. Lack of Orphan Protein Evaluation:
Another potential weakness is that while the method is naturally suitable for orphan proteins, since it relies on a pre-trained language model rather than MSAs, the paper does not provide experiments specifically evaluating its performance on orphan proteins. This omission means that its effectiveness in such scenarios remains unverified on targeted benchmarks.

---

> ### Author Response · Authors · 2025-03-07
> **Author response**
>
> We thank the reviewer for their thoughtful comments. Please see our response to your points below.
>
> > Accuracy Trade-offs: Although MiniFold performs well on many datasets, its accuracy, particularly on the CASP14 dataset and in full-atom structure prediction, is inferior to that of more mature methods such as AlphaFold2 or RosettaFold2. This suggests that the architectural simplifications might compromise the ability to capture fine-grained structural details in challenging cases.
> > Dependence on Pre-trained PLMs: While replacing MSA with a pre-trained language model saves significant time, it may also limit performance in cases where deep evolutionary information (typically captured via MSA) plays a critical role in accurate structure prediction. This reliance might result in diminished performance for proteins where evolutionary context is paramount.
>
> In many applications that require large scale screening, it is acceptable to trade off speed for some accuracy loss. This is evident from the broad use of ESMFold in the field, for example in filtering de-novo protein designs where efficiency is highly desirable [1,2]. It also makes retraining and fine tuning possible with limited resources. With that in mind, the main message of our paper is that the architecture of pLM based structure prediction can be vastly optimized with little to no loss in accuracy. As such, the main point of comparison for our method is ESMFold. The fact that pLM based models perform worse than their MSA counterparts is a well known result from the ESMFold work, since both ESMFold and AlphaFold2 use the same architecture [3]. Instead, our goal was to produce the fastest pLM based model possible. We do agree that for settings where high precision is required, one could combine MiniFold and AlphaFold in subsequent filtering stages to achieve a good balance. It’s also worth noting that some of our work, for instance the inference kernels, can be readily applied to AlphaFold2 as well. We have added some text in section 4.1.1 to include more discussion around these points.
>
> >Lack of Orphan Protein Evaluation: Another potential weakness is that while the method is naturally suitable for orphan proteins, since it relies on a pre-trained language model rather than MSAs, the paper does not provide experiments specifically evaluating its performance on orphan proteins. This omission means that its effectiveness in such scenarios remains unverified on targeted benchmarks.
>
> Thank you for this suggestion, we have added a figure in appendix G showing the model performance as a function of MSA depth for the different datasets, which shows that our model is competitive with ESMFold in all regimes of homology depth, including orphan proteins with little to no homologs.
>
> [1] Campbell, A., Yim, J., Barzilay, R., Rainforth, T. &amp; Jaakkola, T.. (2024). Generative Flows on Discrete State-Spaces: Enabling Multimodal Flows with Applications to Protein Co-Design. Proceedings of the 41st International Conference on Machine Learning
>
> [2] Watson, J.L., Juergens, D., Bennett, N.R. et al. De novo design of protein structure and function with RFdiffusion. Nature 620, 1089–1100 (2023). https://doi.org/10.1038/s41586-023-06415-8
>
> [3] Zeming Lin et al. ,Evolutionary-scale prediction of atomic-level protein structure with a language model.Science379,1123-1130(2023).DOI:10.1126/science.ade2574

---

> > ### Comment · Reviewer_iLp6 · 2025-03-25
> >
> > Thank you for the response, the additional experiment addressed my concern.

---

### Review · Reviewer_3CjT · 2025-02-23

**Summary Of Contributions:**

In this paper the authors introduce MiniFold, an optimized protein structure prediction method that uses protein language models, novel GPU kernels, and ablations to construct an optimized prediction method, without requiring MSAs. They redesign the Evoformer module and adapt the structure module into a simpler, faster alternative, achieving up to 20x speed ups in training and inference. MiniFold is competitive with ESMFold, AlphaFold, and other methods on the CAMEO and CASP benchmarks.

**Audience:**

Yes

**Claims And Evidence:**

Yes

**Requested Changes:**

Thorough responses to the two above questions would secure my recommendation for acceptance.

**Strengths And Weaknesses:**

Strengths

MiniFold is substantially faster at inference time than ESMFold, the most widely used single-sequence structure prediction method. For high-throughput applications like filtering and assessing computational protein designs, these speed increases are very practically useful. The performance is comparable to ESMFold, and the tradeoff in quality vs MSA-reliant methods like AlphaFold is well discussed in the literature and it is clear that for many applications, this tradeoff is acceptable given the significantly faster inference time. So I do not consider the weaker performance compared to AlphaFold as a “weakness” of the method; rather it is a more nuanced tradeoff that practitioners should take into account. And I believe the authors have done a nice job acknowledging this tradeoff and the limitations of their method, which is much appreciated.

Weaknesses

My two main questions / weaknesses relate to the choice of Triton kernels and the availability of the method.
1. Can the authors expand on why they chose to use Triton? How do the advantages and disadvantages of the framework compare to the more commonly used CUDA framework, specifically with respect to structure prediction methods? Are there any concerns about usability or interoperability because of the Triton kernels? Are there plans to develop and release a CUDA-based version of MiniFold, or are there compelling reasons not to do so?
2. Because MiniFold is a structure prediction tool, the claims in the paper must be supported by some usable code. Using a service like https://anonymous.4open.science/, can the authors make at least some portion of the code available for review, and allow for verifying the usability of the model?

---

> ### Author Response · Authors · 2025-03-07
> **Author response**
>
> We thank the reviewer for their thoughtful feedback and for appreciating the merits of our work. Please see the answer to your questions below.
>
> > Can the authors expand on why they chose to use Triton? How do the advantages and disadvantages of the framework compare to the more commonly used CUDA framework, specifically with respect to structure prediction methods? Are there any concerns about usability or interoperability because of the Triton kernels? Are there plans to develop and release a CUDA-based version of MiniFold, or are there compelling reasons not to do so?
>
> We opted for Triton due to its approachability and extensibility. One of the great successes of the Triton project was to enable a broader community of researchers to develop powerful kernels for machine learning models and abstract away many of the intricacies of CUDA. While it may be possible to write a CUDA kernel of slightly better performance, Triton makes our kernels easy to read and understand, allowing us to demonstrate that the key insight behind them is the reduction of memory bound operations. Pytorch itself has transitioned to Triton for many of its operations. Flash-attention has also recently been reimplemented in Triton. As the popularity of the framework continues to grow we do not believe that the lack of direct CUDA implementation will hinder its use but we certainly appreciate the possibility that the Triton compiler could be sub-optimal.
>
> > Because MiniFold is a structure prediction tool, the claims in the paper must be supported by some usable code. Using a service like https://anonymous.4open.science/, can the authors make at least some portion of the code available for review, and allow for verifying the usability of the model?
>
> We actually submitted our code alongside the pdf in the original submission. Please see the supplementary material zip file for our full code base, including our Triton kernel implementations.

---

### Decision · Action_Editor_UNZC · 2025-03-25

**Recommendation:** Accept as is

**Comment:**

The reviewers and this action editor all appreciate this contribution and the authors effort to give the research community access to more efficient protein structure prediction. It can be expected that this paper will have a substantial impact and help spur further progress in accelerating computation in deep learning for point cloud data.

**Audience:**

This paper will have a big audience because protein structure prediction is an important topic and progress in terms of inference speed is important for scaling for example to structure prediction for proteomes.

**Claims And Evidence:**

Yes. Authors have submitted evidence and code to support their claims.